# The Diagnostic Potential of the Human Blood Microbiome: Are We Dreaming or Awake?

**DOI:** 10.3390/ijms241310422

**Published:** 2023-06-21

**Authors:** Francesca Sciarra, Edoardo Franceschini, Federica Campolo, Mary Anna Venneri

**Affiliations:** Department of Experimental Medicine, Sapienza University of Rome, 00185 Rome, Italy; francesca.sciarra@uniroma1.it (F.S.); edoardo.franceschini@uniroma1.it (E.F.); federica.campolo@uniroma1.it (F.C.)

**Keywords:** human blood microbiome, bacteria, dysbiosis, prognostic marker, microbiome

## Abstract

Human blood has historically been considered a sterile environment. Recently, a thriving microbiome dominated by Firmicutes, Actinobacteria, Proteobacteria, and Bacteroidetes phyla was detected in healthy blood. The localization of these microbes is restricted to some blood cell populations, particularly the peripheral blood mononuclear cells and erythrocytes. It was hypothesized that the blood microbiome originates from the skin–oral–gut axis. In addition, many studies have evaluated the potential of blood microbiome dysbiosis as a prognostic marker in cardiovascular diseases, cirrhosis, severe liver fibrosis, severe acute pancreatitis, type 2 diabetes, and chronic kidney diseases. The present review aims to summarize current findings and most recent evidence in the field.

## 1. Introduction

Traditionally, some body compartments such as the upper female gynecological tract [1], brain [2], and blood [3] were considered microbiologically sterile in healthy conditions. However, several works pointed to how these and other body niches cannot be longer considered germ-free in healthy conditions due to the presence of commensal or native microbial flora (collectively named the microbiome) [3]. The existence of metabolically active bacteria in the blood was hypothesized for the first time 50 years ago following the detection of increased absorption of amino acids and nucleosides in the erythrocyte suspension of healthy individuals [4]. Bacterial growth was also confirmed, following osmotic lysis and filtering, in healthy blood samples [5]. Furthermore, the advent of advanced analytical approaches, such as real-time PCR, allowed for the detection of ribosomal DNA in the same samples [6]. Moreover, the use of dark field microscopy techniques, fluorescent in situ hybridization, and flow cytometry led to the identification of pleomorphic bacteria in the blood, capable of altering their characteristics in response to environmental conditions [7]. Moreover, the analysis of blood microbiome morphology in peripheral blood mononuclear cells (PBMCs) using light and electron microscopy highlighted how the blood microbiome displays a complex life cycle, undergoing different morphological transformations, such as vesiculation, tabulation, budding, irregular binary fission, and protrusion–extrusion of progeny cells from large electron-dense bodies [8]. Recently, several studies have managed to increase the “power” of analysis due to a lower abundance of bacterial DNA, investigating healthy blood microbial taxonomy, identifying the most abundant phyla, and sometimes also bacterial genera [9] using 16 s rRNA gene sequencing technology [10,11]. In some blood compartments (red/white blood cells, extracellular vesicles) specific bacteria were detected. Their cellular localization (intracellular, membrane-bound) was linked to specific bacterial behavior (intracellular parasite, commensal), which in turn can depend on environmental conditions [12].

Moreover, blood microbiome sequencing studies highlighted how bacterial abundance differs among European countries. Bacterial DNA abundance distribution was differentially detected according to geographical origin: Germany and Poland show significantly higher abundance values; Italy, Greece, and Finland present intermediate levels, while Belgium and Austria are characterized by lower blood bacterial DNA levels. The main differences among countries were observed in male subjects, together with no age-related differences in bacterial DNA amount detected. This phenomenon could be at least in part attributable to environmental, behavioral, and cultural factors and social habits [13]. Some microbiome-characteristic variations were also correlated with host genetic and innate/adaptive immunity differences, while in other cases, cultural/behavioral features such as diet, hygienic conditions, parasitic load, and environmental exposure appear to have major weight compared to genetic factors [14]. Moreover, an age-based microbiome property difference was not highlighted, suggesting how the environment could have a greater contribution than age [15] even if an association between the blood microbiome and older age cannot be entirely excluded. Indeed, intestinal dysbiosis, malabsorption, and immune response dysregulation are common features of older subjects and lead to chronic systemic inflammatory states [16].

Even if origin and difference in the microbiome have been highlighted, many questions still remain unanswered, the first of which regards the blood microbiome’s source. It is unclear if it comes from other compartments, such as the gut [17], or if it is native to the blood environment [18]. Although members of the microbiota are often referred to as commensals, the symbiosis—persistent interaction—between the microbiota and its mammalian host encompasses various forms of relationship including mutualistic, parasitic, or commensal.

Gut microbiome–host interaction can be highly contextual, with the same microbe developing as a mutualist or parasite according to the nutritional, coinfection, or genetic landscape of its host. Many studies have examined the correlation between the gut microbiota and the immune system by analyzing its role in the development and promotion of autoimmune diseases and several cancer types [19,20].

Moreover, the persistence of blood microbes can lead to the development of many diseases including liver–pancreatic pathologies or cardiovascular and kidney diseases [21,22], although not always necessarily due to an immunomodulatory capacity of circulating commensals.

The bloodstream allows microbes to reach different body sites in healthy individuals; however, the low prevalence of such microbes in healthy blood underscores the transient and infrequent nature of this phenomenon [23].

The aim of this manuscript is to deeply review the relationship among bloodborne bacteria dysbiosis and several human disorders, highlighting the clinical relevance of changes in blood microbiome composition, and to unlock the potential microbiota-derived biomarkers for the diagnosis of a variety of diseases.

## 2. Blood Microbiome Composition

A healthy gut microbiome is mainly characterized by Firmicutes and Bacteroidetes phyla domination [24]. Knowledge on the healthy blood microbiome is still an under-evaluated topic, even though many research groups focus on this issue. Results from these analyses converge on a microbiome dominated by Firmicutes, Actinobacteria, Proteobacteria, and Bacteroidetes phyla (Table 1) [9,25], with less abundant Fusobacteriota [26,27]. Some genera belonging to the phylum Firmicutes were identified in blood: *Bacillus*, obliged aerobic bacteria widespread in nature; [9,26,28,29] *Lactobacillus,* facultative anaerobic able to convert lactose and other sugars in lactic acid by lactic fermentation; *Streptococcus*, facultative anaerobic able to produce hemolytic exotoxins such as streptolysin; and *Romboutsia* [30], a gut species isolated in humans [31]. Moreover, the Actinobacteria genus *Corynebacterium*, commensal human bacteria usually not pathogenic in healthy state; Proteobacteria genus *Pseudomonas*, bacteria with great metabolic diversity able to colonize a wide range of niches [9]; and *Bacteroides*, a Bacteroidota genus associated with the gut microbiome, able to process complex polysaccharides into simpler ones were detected in healthy blood [30,32]. The microbial DNA detected in healthy subjects comes from circulating commensals that have asymptomatic coexistence with the host and display immunomodulatory phenotypes. The presence or absence of immunomodulatory phenotypes of the microbiome determines whether an individual with bacteremia is asymptomatic or septic [23].

## 3. Blood Microbiome Origins

Many hypotheses have been postulated regarding the human blood microbiome’s origin, making this issue a question still unsolved. A recent population study rejected the hypothesis of an endogenous blood microbiome, rather supporting the transient and sporadic translocation of commensal microbes from other body sites into the bloodstream [23]. Other works have instead considered such translocations as a frequent event in healthy conditions, frequent enough to support the existence of a stable blood microbial microbiome [37]. In addition, some works have pointed to a bacterial contamination from the skin, laboratory environments, or other environmental sources, discrediting the idea of an endogenous blood microbiome [26,37]. It the origin of the healthy microbiome has also been hypothesized, and it appears that bloodborne microorganisms, especially bacteria, may have a maternal origin [38]. However, based on taxonomy detection, some suitable bacterial origin niches were identified. Several studies detected most of the blood microbiome in the gastrointestinal tract [39] even if it presents marked taxonomic differences compared to the gut microbiome [40]. Other sites of origin are the skin and oral mucosa, where blood bacterial phyla were detected [10,41]. The intestinal immune system is exposed to a number of foreign antigens, coming both from diet and from commensal or pathogenic bacteria presence; thus, the gut mucosa could represent a bacterial route toward the bloodstream. Different bacterial translocation mechanisms across the blood barrier have been hypothesized. One of them involves dendritic cells (DCs). The lamina propria (LP) mucosae contains various populations of DCs, such as CD103(+) and CX(3) CR1(−). DC antigen transport could represent a bacterial translocation way through the intestinal barrier [42]. In addition, other cell types could play a supporting role for intestinal DCs: small intestine mucus-secreting goblet cells (GCs) were found to be able to deliver small molecular antigens from intestinal lumen to CD103(+) DCs [43]. Moreover, intestinal microfold cells (M cells), specialized epithelial cells of the mucosa-associated lymphoid tissues, can sample various antigens and move them, therefore representing a pathway for the intestinal microbiome and antigens [44]. For skin and oral mucosa sources, bacterial translocation may occur when the blood barriers are blocked, as in the case of *Porphyromonas gingivalis* bacteremia [45].

Maternal origin is an alternative hypothesis to explain the source of the blood microbiome: the vertical bacterial transmission is indeed a phenomenon already known in animals for gut microbiome bacteria [46]. Microbiome presence was detected in various pre-natal tissues, such as umbilical cord blood [47], meconium [48], amnion [49], and in the placenta, which harbors a low-biomass microbiome [50]. All such evidence, although controversial, [1,51] could also point to theoretical maternal bacterial translocation in humans. The mechanisms by which maternal bacterial transfer would work are still unknown, even if some studies suggest a mixed origin, before and after birth. The initial microbial inoculum would be maternal, by oral and gut fetal compartment colonization, and the microbial microbiome would then be enriched during nursing [52]. Another hypothesized bacterial translation way could be the fetal ingestion of amniotic fluid during gestation [53]. Taking into account all these considerations, further investigation is needed to deepen understanding of the blood microbiome’s origin.

## 4. Blood Microbiome Localization

The healthy blood microbiome has been localized in many human blood sources: in the buffy coat (BC); inside peripheral blood mononuclear cells (PBMCs), such as commensal bacteria associated with red blood cells; and in red blood extracellular vehicles (EVs). The specific blood cell localization reflects bacterial behavior: an obliged intracellular parasite exploits internal cells [12], commensal species use tactics to stick to them [28], while opportunistic bacteria can change their habits depending on the environmental conditions [54]. Most bacterial DNA was detected in the blood buffy coat in healthy individuals [27]. In a significant percentage of healthy donors, PBMC phylum Chlamydiae species *Chlamydophila pneumoniae*, a human obligate intracellular parasite bacterium and etiologic agent of an atypical pneumonia form [55], was detected (Table 2). Chlamydiae are known for their unique biphasic life cycle, during which they alternate between two morphological forms: infectious extracellular elementary bodies (EBs) transform into metabolically active intracellular reticulate bodies (RBs) after entering the host cell [12].

### 4.1. Commensal Bacteria

In various blood compounds, phylum Firmicutes species *Staphylococcus aureus*, a broadly diffused commensal bacterium that can shift to opportunistic pathogen state depending on environmental conditions, was detected, representing one of the most common bacteremia causes, also infecting skin and soft-tissues [54] (Table 2). It represents a versatile bacterial species that can display a relevant number of strategies to exploit host cells [56], such as to being phagocytized by white blood cell neutrophils and macrophages. Some bacterial strains can survive inside white blood cells, deactivating their antibodies, and using them to facilitate bacterial dissemination to other compartments [57]. In addition, *Staphylococcus aureus* can also exploit red blood cells: it needs iron uptake from the host for its activity and uses complex strategies to gain it. The *Staphylococcus aureus* heme molecule uptake system uses highly specific cell surface receptors to extract iron ion from hemoglobin and hemoglobin–haptoglobin complexes, to transport it into the cytoplasm, and to degraded it as free iron [58,59] (Table 2). Another commensal phylum Firmicutes species, *Streptococcus pneumoniae*, manipulates red blood cells and uses them to evade innate immunity. The mechanism by which it works is still under investigation, but recent data point to surface protein PfbA’s (plasmin-fibronectin-binding protein A) role in erythrocytes binding due to its relevant affinity toward red blood cells and moderate affinity toward hemoglobin [60]. *Streptococcus pneumoniae* is involved in lobar pneumonia disease, giving rise to an inflammatory process in the alveolar spaces, with the exudation of a protein-rich liquid. This fluid works as a bacterial culture medium for dissemination, typically causing pneumonia [61].

**Table 2 ijms-24-10422-t002:** Localization of bacteria in the blood and identification of their behavior.

Phyla	Species	Localization	Clinical Effects	Bacterial Behavior	References
Chlamydiae	*Chlamydophila pneumoniae*	PBMCs	Etiologic agent of an atypical pneumonia form	Obligate intracellular parasite	[55]
					[12]
					[62]
Firmicutes	*Staphylococcus aureus*	White blood cells	Cause bacteremia	Commensal/ opportunistic pathogen	[56]
					[57]
		Red blood cells	Extract iron ion from hemoglobin and hemoglobin–haptoglobin complexes		[58]
	*Streptococcus pneumoniae*		Involved in lobar pneumonia disease onset		[60]
					[61]
	*Staphylococcus epidermidis*	Plasma and Red blood cells	Involved in nosocomial infections of immunocompromised and transplanted patients		[28]
					[63]
Proteobacteria	*Serratia* sp.				[29]
Pseudomonadota	*Acinetobacter baumannii*				[37]
	*Stenotrophomonas maltophyla*				
Actinobacteria	*Cutibacterium acnes*	Blood samples	Causes chronic blepharitis and endophthalmitis		[28]

### 4.2. Opportunistic Pathogens

*Staphylococcus epidermidis* acts as a skin commensal bacterium in healthy conditions, but it can switch to opportunistic pathogenic behavior depending on environmental conditions, causing nosocomial infection to immunocompromised or transplanted patients. It was detected as being associated with the red blood cell fraction of healthy donors [28] (Table 2). *Staphylococcus epidermidis* has surface and extracellular matrix proteins that can bind blood cells. With the protein polysaccharide intercellular adhesin (PIA), it can create an extracellular material, a biofilm, which can also bind other bacteria and create a multilayer biofilm [63].

Moreover, phylum Actinobacteria species *Cutibacterium acnes*, a commensal anaerobic bacterium of the human skin microbiome [64], can switch its behavior as an opportunistic pathogen. Some *Cutibacterium acnes* strains produce several virulence factors involved in polysaccharidic biofilm production, adhesion to target cells, inflammation targeting by enzymatic proteins, and host tissue degradation [65]. Due to their ability to persist on medical devices, they were also involved in a wide range of nosocomial infections, and in chronic disease onset, such as endocarditis [64]. *Cutibacterium acnes* was detected in blood of healthy donors together with other opportunistic pathogenic bacteria such as *Staphylococcus epidermidis* [28].

Other environmental opportunistic human pathogens, known to cause nosocomial infections in immunocompromised patients, were also found to be associated with the healthy red blood cell fraction: among them, Proteobacteria genus *Serratia* [29] and Pseudomonadota species *Stenotrophomonas malthophyla* and *Acinetobacter baumannii* [37] (Table 2 and Figure 1). Recently, Proteobacteria, Firmicutes, and Actinobacteria phyla and *Staphylococcus*, *Bacillus*, *Corynebacterium*, *Pseudomonas*, *Acinetobacter*, *Cutibacterium* genera were detected in human femoral arteries [9] (Table 2). Moreover, many indirect clues to bacterial presence in red blood cells of healthy donors’ EVs suggest the presence of lipopolysaccharides (LPS) and outer membrane protein A (OmpA) [66]. In addition, EVs are also able to interact with monocytes, pointing to a microbial crossover among blood cells [66].

## 5. Clinical Relevance of Blood Microbiome Dysbiosis

Dysbiosis is a term that refers to a change in bacterial composition and metabolite production in microbial communities [67]. The use of microbial dysbiosis as a biomarker for human pathology has mainly been studied for the human gut microbiome [17]. Some recent discoveries, such as a significant positive association between blood microbial DNA copy number and important factors for health, such as glucose, insulin, free fatty acid (FFA) levels, and leukocyte count, led researchers to hypothesize the use of blood microbiome dysbiosis as a potential prognostic biomarker for several human pathologies [13]. However, it should be noticed that specific blood phyla in diseased patients are not indicative of bacterial origin site or of some specific causative role in pathology onset.

### 5.1. Blood Microbiome and Gastroenteropancreatic and Pulmonary Diseases

Different works have analyzed the blood microbiomes of patients affected by gastro-enteropancreatic diseases highlighting a dysbiosis status. Cirrhotic patients showed an increase in blood total bacterial DNA concentration and higher bacterial heterogeneity compared to healthy subjects [68,69]. Patients affected by severe liver fibrosis showed unique bacterial taxonomy and higher 16 rRNA concentration compared to healthy subjects [70] and showed a specific taxonomic composition for portal vein and central–peripheral venous blood and liver outflow [71]. In severe acute pancreatitis (SAP), patients’ microbial taxonomic diversity appears reduced compared to healthy subjects, with an increase in phyla Bacteroidetes and Firmicutes and a decrease in Actinobacteria (Table 2). Such microbial variations in SAP patients were associated with several immunological dysfunctions characterized by increased levels of serum cytokines, lymphocyte subgroup disruption, and neutrophil proteomic profile alteration [34]. The bacterial and viral microbiomes were identified in chronic obstructive pulmonary disease (COPD) and a relationship between the systemic microbial populations and lung disease was highlighted [72].

### 5.2. Blood Microbiome and Diabetes

Gut microbiome dysbiosis was associated with diabetes onset [36,73,74] and a link between gut bacterial dysbiosis and intestinal bacteria translocation to bloodstream was also proposed, as diabetic patients’ blood presented a higher rate of gut-derived microbiome than healthy controls [75]. Individuals with type 1 diabetes of long duration and with initial stages of diabetic nephropathy are characterized by aberrant profiles of gut microbiota and plasma metabolites [74]. A case–control study on insulin resistance syndrome highlighted how people predisposed to developing diabetes showed higher 16 S rDNA concentration in their blood compared to people not predisposed to developing the disease [76]. Other works did not detect bacterial dysbiosis in T2D patients, even though study participants who carried the *Bacteroides* genus in their blood were significantly associated with a lesser risk of developing T2D, while the *Sediminibacterium* genus bearers showed higher risk of becoming diabetics [36]. Thus, blood *Bacteroides* and *Sediminibacterium* could also be considered as T2D biomarkers in the future, although more study is needed.

### 5.3. Blood Microbiome and Acute Cardiovascular Disease Patients (CVD)

Acute cardiovascular disease (CVD) patients showed a relevant decrease in blood bacterial DNA compared to healthy subjects, and, in addition, patients developing cardiovascular complications showed a nonsignificant increase in Proteobacteria phyla. Proteobacteria augmentation could thus represent an independent biomarker of cardiovascular disease [21]. CVD patients’ blood dysbiotic state has been confirmed by other studies: CVD blood showed a dominance of Actinobacteria and Proteobacteria phyla compared to healthy blood [33]. In addition, an increased proportion of Proteobacteria and Actinobacteria was also detected during cardiac events, such as stroke, coronary heart disease, and myocardial infarction, confirming the proposed role of independent biomarker for Proteobacteria and Actinobacteria in CVD risk [25]. Proteobacteria species *Escherichia coli*, a common commensal bacterium harbored in the human gut, was detected in the blood of myocardial infarction (MI) coronary thrombosis patients [77]. In particular, the *Escherichia coli* LPS strain found in MI blood showed an association with several pathological factors involved in MI disease: low-grade endotoxemia, which triggers an inflammatory state characterized by proinflammatory cytokine release; and with soluble P-selectin protein, a marker of platelet activation [78,79]. In addition, *Escherichia coli* LPS strain also showed significant association with zonulin, a protein marker of gut permeability, also pointing to a factor involved in the bacterial translational pathway from the gut to blood circulation [78]. The role of Proteobacteria phyla in the pathogenesis of cardiovascular diseases was explored, detecting a link between LPS, a relevant Proteobacteria component, and CVD onset [25]. LPS can trigger inflammation directly, by its microbe-associated molecular patterns, or indirectly, inducing the generation of nonmicrobial danger-associated molecular patterns. Translocation of LPS in blood causes endotoxemia, and chronic endotoxemia is in turn involved in the pathogenesis of inflammation-driven conditions such as cardiometabolic disorders (obesity, liver diseases, atherosclerotic cardiovascular diseases) [80]. Moreover, the association between blood bacterial DNA and the risk of cardiovascular mortality was analyzed, detecting three bacterial genera significantly associated with it. Phylum Actinobacteria genus *Kokuria* and Pseudomonadota genus *Enhydrobacter*, skin and oral bacteria, were identified as being directly associated with CVD mortality risk, while Pseudomonadota genus *Paracoccus* was found to be inversely correlated (Table 2) [81]. Such findings highlight a role for blood microbiome dysbiosis in cardiovascular diseases, even though it should be considered that a clinical association between CVD and blood dysbiosis is still inconsistent, and more studies are needed to deepen knowledge on a theoretical causative role of bacteria in cardiovascular diseases.

### 5.4. Microbiome and Chronic Kidney Disease Patients (CKD)

A pilot study examined circulating microbiome profiles of chronic kidney disease (CKD) patients and compared them with healthy controls. A significant reduction in bacterial diversity in CKD blood and an augmentation of phylum Proteobacteria (Enterobacteridaceae and Pseudomonadaceae families) [22] (Table 2) were detected. The Proteobacteria phylum is a relevant factor in CKD pathogenesis due to its bacterial LPS. High LPS levels in the blood, together with gut-derived uremic toxins and immune deregulation, played a critical role both in CKD and CKD-associated complications [82]. Other studies confirmed CKD blood dysbiosis by the detection of lower bacterial heterogeneity and significant taxonomic variations [83]. In addition, a pilot study explored the link between the gut and blood microbiome in CKD patients with or without vascular calcification (VC), detecting significant changes in specific gut and blood phyla [84].

### 5.5. Microbiome and Cancer

Unique blood microbiome profiles have recently been identified in several cancer types and whole genome transcriptome studies have shown how unmapped parts of the reads can be frequently assigned to microorganisms, such as bacteria, archaea, and also viruses [85]. Despite intratumor microbial signatures being detected as effective in normal tissue discrimination from tumors, they did not correlate with cancer stages in most of the cancer types [85]. Instead, blood microbial profiles of cancer patients were sensitive enough to detect different types of cancer, even for very early-stage cancers. Such findings point to a blood/intratumor microbiome-based diagnostic method, which could be helpful in the screening and early diagnosis of cancer [18]. In addition, blood bacterial genetics showed potential regarding cancer treatment response prediction: in advanced colorectal cancer, baseline blood microbiota composition was identified as being significantly different among immunochemotherapy responders and non-responders [86].

Relevant findings were also gained studying the potential impact of the blood microbiome on immunotherapy efficacy [86]. Circulating microbiome effect on immunotherapy efficacy was checked on advanced colon/colorectal cancer (CRC) patients subjected to adoptive cellular therapy (ACT) with a mixed dendritic cell/cytokine-induced killer cell product (DC/CIK). The clinical benefits of DC/CIK infusion resulted in being influenced by blood microbiome diversity. DC/CIK responders, which had better immunotherapy impact compared to non-responders, experienced blood microbiome changes, with an enrichment of Actinobacteria genus *Bifidobacterium*, and Firmicutes genera *Lactobacillus* and *Enterococcus* [86]. Some studies on murine models showed how *Bifidobacterium*, *Lactobacillus*, and *Enterococcus* strains showed a promoting impact on immunotherapy effects [87]. *Lactobacillus rhamnosus* GG oral administration to murine models was connected to an augmentation of immunotherapy anti-programmed cell death 1 (PD-1) antitumor activity, by T-cell and tumor-infiltrating DC increase [88], supporting the potential use of *Lactobacillus rhamnosus* GG in human CRC patients to raise their antitumor response and immunotherapy efficacy [89]. Several *Bifidobacterium longum* strains showed antitumor properties in tumor-bearing mice, used alone or in combination with cyclophosphamide (CTX), raising leukocytes levels and significantly inhibiting mouse tumor growth [90]. Some species of the *Enterococcus* genus, such as *Enterococcus faecalis*, were found to be connected with an improvement in checkpoint inhibitor immunotherapy in tumor-bearing mice. *Enterococcus* can express the SagA (Spt–Ada–Gcn5–acetyltransferase) protein, a transcriptional coactivator complex, able to induce the production of immune-active muropeptides, peptydoglican molecules associated with cell walls. The expression of SagA by *Enterococcus faecalis* was detected, connected to an immunotherapy enhancement in tumor-bearing mice [91]. Even if such evidence is restricted to preclinical models, it clearly suggests how the blood microbiome can exert an effect on immunotherapy clinical response, laying the foundations for considering the blood microbiome as a potential immunotherapy biomarker [87].

### 5.6. Microbiome and Fertility

Blood microbiome dysbiosis has also been detected in polycystic ovary syndrome (PCOS): the relative abundance of Proteobacteria, Firmicutes, and Bacteroidetes decreased significantly, while Actinobacteria increased significantly (Table 2) [35]. Many studies also point to the potential roles of the gut microbiome in hormonal alteration related to the pathogenesis of PCOS [92,93]. Moreover, blood microbiome composition undergoes a significant alteration in PCOS patients compared to healthy subjects [90]. Variations in the blood microbiome and the co-occurrence of harmful pathogens with other bacteria, fungi, and viruses may play a role in pregnancy-related complications, and their study could be useful to stratify high/low-risk patients [18]. However, up to now, no data on a connection between circulating blood microbiota dysbiosis and male fertility have been produced.

## 6. Microbial Metabolites Identification in Disease

Several sophisticated techniques, such as next generation sequencing, mass spectrometry, and biosensors/nanosensors, are currently used to analyze circulating microbiomes and their metabolites. Mass spectrometry-based metabolomics constitutes an innovative approach to the detection and characterization of small molecules [94] and microbiome-synthesized metabolites [95]. Furthermore, the knowledge of a specific metabolomic signature represents relevant information that allows for the identification of bacterial sources [96]. Moreover, comparing the blood metabolome after a pharmacologic treatment allows for detecting changes in the bacterial microbiome. Such a bacterial shift could point to an effect of the pharmacological treatment and may also lead to a better understanding of microbial dynamics [97]. The analysis of circulating metabolites in obese patients highlighted a change in the abundance of the Bacteroidota species *Bacteroides thetaiotaomicron* compared to healthy controls [98]. Such bacterium is an obligate anaerobic species, a common human gut commensal–opportunistic pathogen, whose genome contains a relevant number of polysaccharide digestion-associated genes, making it able to hydrolyze most biological glycosidic bonds [99]. In addition, *Bacteroides thetaiotaomicron* enacts a decarboxylase action on glutamate, a neurotransmitter, converting it into γ-aminobutyric acid, which has a relevant role in in gut–brain axis homeostasis [100]. Moreover, obese patients display an inverse correlation between blood bacterial load and glutamate concentration. A bariatric surgery weight-loss intervention reversed microbial and metabolic obesity-connected disruptions, such as *Bacteroides thetaiotaomicron* decrease and a rise in serum glutamate. Such data highlighted a link between gut microbiome metabolic secretion disruption and obesity, also suggesting the use of the serum *Bacteroides thetaiotaomicron* dysbiosis as an obesity biomarker [98]. A study analyzing major depression-affected patient blood microbiomes revealed that affected patients showed a distinctive blood microbiome compared to controls, and metabolomic profiling showed how diseased patients had alterations in their cyanoaminoacid pathways. After pharmacological treatment, an augmentation of Firmicutes associated with a decrease in Actinobacteria and Pseudomonadota phyla was associated with response to antidepressant treatment. Based on detected metagenomic profiles, bacterial profiles that resulted in association with treatment response were those related with amino acids, lipids, carbohydrates, tryptophan, and drug metabolism [101]. As previously discussed, blood microbiome and metabolomic signature information are relevant to understanding the prediction of risk factor in coronary artery disease (CAD)-affected patients with a distinct serum metabolome compared with control individuals. The Clostridiaceae bacterial family was associated with levels of circulating metabolites, several of which have previously been linked to an increased risk of CAD [102]. These results highlight the powerful utility of the serum metabolome in understanding the basis of risk factor heterogeneity in CAD. All studies have shown that bacterial metabolome detection could represent a useful diagnostic tool; however, further dedicated studies are needed.

## 7. Final Remarks

Based on the data examined, the microbial presence of the bacterial microbiome in healthy blood is a hot topic in favor of which a still-increasing number of evidence is being collected. All studies focus on circulating microbiome bacterial characterization, on their cellular localization, and on understanding the relationship between the blood microbiome and the development of pathological diseases. Unlike a healthy gut, which is known to be characterized by Firmicutes and Bacteroidetes phyla, many works addressed healthy blood microbiomes dominated by Proteobacteria, Actinobacteria, Firmicutes, and Bacteroidetes. Actually, a few specific blood compounds where healthy a blood microbiome is harbored were localized: in buffy coat; inside peripheral blood mononuclear cells, such as commensal bacteria associated with red blood cells; and in red blood extracellular vesicles.

The immune system is shaped by the human microbiome. It is known that the gut microbiota and immune cells interact with each other through several mechanisms, including the engagement of Toll-like receptors performing immunomodulation, actively impacting multiple host function including circadian rhythmicity, nutritional responses, and metabolism. Perturbation of the microbiome impairing the host–microbiome interface or an altered immune system can result in systemic dissemination of commensal bacteria, susceptibility to infections, and abnormal immune responses.

The identification of potential biomarkers is considered a challenging task for many diseases. Blood microbiome analysis can be considered a noninvasive approach to identifying disease biomarkers and may increase the accuracy of disease classification and the efficacy of therapies. Nevertheless, many unresolved questions remain, such as disentangling microbiome–immune system binomial in homeostasis and disease.

The present review deeply dissected the physiopathology of the blood microbiome from healthy blood microbiome composition, origins, and localization to blood microbiome dysbiosis and its crosstalk with various diseases.

Due to the novelty of the topic, further stringent preclinical and clinical studies are needed to identify the circulating microbiome both in terms of characterization and function, revealing the molecular mechanisms of microbial translocation, and their role in the onset of diseases. Furthermore, this overview could be useful to develop personalized diagnostic and therapeutic microbiome-targeted treatment.

## Figures and Tables

**Figure 1 ijms-24-10422-f001:**
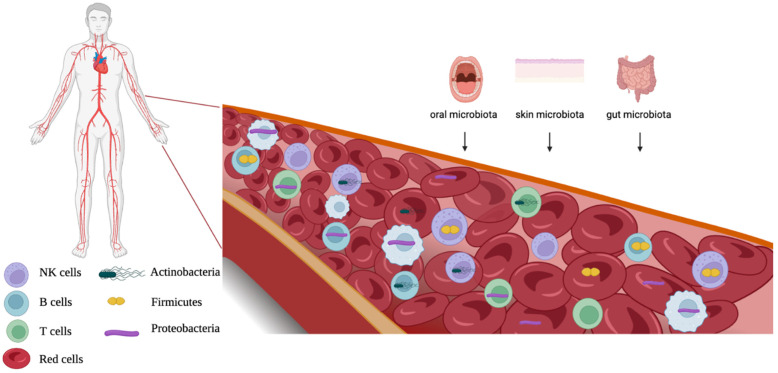
Origins and localization of bacteria in blood cells.

**Table 1 ijms-24-10422-t001:** Identification of blood microbiome composition in the healthy and diseased.

Condition	Phyla	References
Healthy	Actinobacteria Firmicutes Proteobacteria Bacteroidetes	[9,25]
CVD	Actinobacteria Proteobacteria	[33]
SAP	Bacteroidetes Firmicutes	[34]
CKD	Proteobacteria	[22]
PCOS	Proteobacteria Firmicutes Bacteroidetes	[35]
T2D	Bacteroidetes	[36]
Nosocomial Infection	Proteobacteria Pseudomonadota	[29,37]

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
