# Peer review of "The Diagnostic Potential of the Human Blood Microbiome: Are We Dreaming or Awake?"

_ijms, 2023, doi:10.3390/ijms241310422_

Round 1
Reviewer 1 Report
The article is consistent within itself. The references are relevant and recent. The cited sources are referenced correctly. Appropriate and key studies are included. The paper is comprehensive, the flow is logical and the data is presented critically.
However, there are some specific comments on weaknesses of the article and what could be improved:
Major points - none
Minor points
1. Some sections are too long (i.e., 4) - it would be more comprehensive to use subheadings in it for every 250-300 words
2. There is no clear goal/hypothesis in the introduction for the aim of the review; similarly, the conclusions are not strong enough.
Reviewer 2 Report
Sciarra et al addressed a relevant topic of blood microbiome composition and its potential role as a biomarker of different diseases. This review discusses healthy blood microbiome composition, blood microbiome origins and localisation, blood microbiome dysbiosis and its crosstalk with various diseases. The topic of the review is definitely important and is of interest for scientific community. However, there are following comments to be addressed.
1. There are recent reviews published on the same topic, such as e.g. https://doi.org/10.3389/fcimb.2019.00148, https://doi.org/10.3390/ijms24065633. They also discuss healthy blood microbiome composition, blood microbiome dysbiosis and its crosstalk with various diseases, blood microbiome origins and localisation. The authors should consider stating what additional value does their review add to the existing knowledge.
2. The authors should consider adding the information on crosstalk between blood microbiome and immune system. The fact that bacteria are constantly present in the circulation and even localise in the circulating immune cells should have an impact on the immune cells function. If there is not enough literature available on that topic, the authors could discuss what are the possible consequences of such interaction.
3. A table or figure would be beneficial to visualise the associations of blood microbiome compositions and various diseases.
4. The authors should consider rephrasing the lines 307-309, 328-329, as they are not fully clear.
5. The authors should consider revising the part 6. Blood microbiome analysis, as the content of this part does not fully correspond to the title - it mostly discusses the bacterial metabolites in the blood and its association with different diseases.
6. The authors should consider adding a table or a figure to visualise the blood microbiome composition in health and various diseases.
The authors should consider a proof-reading of their manuscript, as there are typos and English grammar mistakes.
Reviewer 3 Report
Interesting review paper on the relationship between blood-borne bacteria, dysbiosis and the development of various human disorders.
Substantiated description of the composition, origins, location, dysbiosis and analysis of the blood microbiome.
High number of recent bibliographical references, adequate to the content of the paper.
Reviewer 4 Report
The review is devoted to the topic of the microbiome of human blood. Until recently, such places as amniotic fluid, blood, and brain were considered sterile and free from bacteria. However, in recent years there has been a literature on the detection of circulating DNA and how the bacteria survives in blood cells. The review is a continuation of the review of Castillo et al., 2019 (Castillo, D. J., Rifkin, R. F., Cowan, D. A., & Potgieter, M. (2019). The healthy human blood microbiome: fact or fiction?. Frontiers in cellular and infection microbiology, 9, 148.), and the work included articles from 2020-2022 with high theoretical significance and can be recommended for publication.
Minor comments
throughout the text (lines 8, 23, 71, 73, 92, 95, 121, 212, 283) – replace "fauna" with "flora" or "microbiome"
line 48 – "characteroized" replace "characterized"
line 249 – "Psudomonadota" replace "Pseudomonadota "
line 256 – " Enteobacteridaceae " replace " Enterobacteridaceae "
lines 44-60 – reference numbering should be continuous (references 52, 84, 85, 86)
line 314 – reference numbering should be continuous (start from 84 but 87)
rewrite Author Contributions according CRediT
Reviewer 5 Report
The current manuscript focused on the blood microbiome, and reviewed the composition, origins, localization, dysbiosis and analysis of blood microbiome. However, verious review had been published among these years, such as “The conserved phylogeny of blood microbiome”, “The dormant blood microbiome in chronic, inflammatory diseases”, “Peripheral Blood Bacterial and Viral Microbiome in COPD”. Thus the current topic lack of innovation and novelty. Otherwise, the main content of current review is too loosely organized, and the topic should focus on a certain type of disease or disease mechanism. In general, I think the review can not be published in current type.
Author Response
Please see the attachment."
